# Exploring the Impact of Water Content in Solvent Systems on Photochemical CO_2_ Reduction Catalyzed by Ruthenium Complexes

**DOI:** 10.3390/molecules29204960

**Published:** 2024-10-20

**Authors:** Yusuke Kuramochi, Masaya Kamiya, Hitoshi Ishida

**Affiliations:** 1Institute of Industrial Science, The University of Tokyo, 4-6-1 Komaba, Meguroku, Tokyo 153-8505, Japan; 2Department of Chemistry, Graduate School of Science, Kitasato University, 1-15-1 Kitasato, Minami-ku, Sagamihara 252-0373, Kanagawa, Japan; 3Department of Chemistry and Materials Engineering, Kansai University, 3-3-35, Yamate-cho, Suita 564-8680, Osaka, Japan

**Keywords:** cage escape, electron transfer, water content, CO_2_ reduction, molecular catalyst, ruthenium complex, bipyridine, photosensitizer, NADH model, unnatural amino acid

## Abstract

To achieve artificial photosynthesis, it is crucial to develop a catalytic system for CO_2_ reduction using water as the electron source. However, photochemical CO_2_ reduction by homogeneous molecular catalysts has predominantly been conducted in organic solvents. This study investigates the impact of water content on catalytic activity in photochemical CO_2_ reduction in *N,N*-dimethylacetamide (DMA), using [Ru(bpy)_3_]^2+^ (bpy: 2,2′-bipyridine) as a photosensitizer, 1-benzyl-1,4-dihydronicotinamide (BNAH) as an electron donor, and two ruthenium diimine carbonyl complexes, [Ru(bpy)_2_(CO)_2_]^2+^ and *trans*(Cl)-[Ru(Ac-**5Bpy**-NHMe)(CO)_2_Cl_2_] (**5Bpy**: 5′-amino-2,2′-bipyridine-5-carboxylic acid), as catalysts. Increasing water content significantly decreased CO and formic acid production. The similar rates of decrease for both catalysts suggest that water primarily affects the formation efficiency of free one-electron-reduced [Ru(bpy)_3_]^2+^, rather than the intrinsic catalytic activity. The reduction in cage-escape efficiency with higher water content underscores the challenges in replacing organic solvents with water in photochemical CO_2_ reduction.

## 1. Introduction

In recent years, technology for CO_2_ reduction has emerged as an effective solution for addressing the global warming problem and the storage of fossil fuels. Metal complexes have become a promising candidate for catalysts in CO_2_ reduction because of their multiple accessible redox states, high activation energies against proton reduction, and various molecular design possibilities through the combination of appropriate metal ions and ligands [1,2,3,4,5,6,7,8,9,10]. Among them, metal diimine carbonyl complexes have been identified as efficient catalysts for selectively yielding CO and/or formic acid without accompanying hydrogen evolution, even though proton reduction is a more thermodynamically favorable process than the CO_2_ reduction. In photochemical CO_2_ reduction, a rhenium diimine carbonyl complex, *fac*-Re(bpy)(CO)_3_Cl (bpy = 2,2′-bipyridine), acts as a highly efficient CO_2_ reduction photocatalyst to selectively produce CO in *N,N*-dimethylformamide (DMF) solution containing triethanolamine (TEOA) as an electron donor [11,12]. Here, *fac*-Re(bpy)(CO)_3_Cl functions not only as the catalyst but also as the photosensitizer. In the case of ruthenium diimine carbonyl and manganese diimine carbonyl complexes, direct irradiation causes the dissociation of the CO ligands [13,14,15]. Therefore, the use of additional photosensitizer is essential. Tris(diimine)ruthenium complexes are often used as the photosensitizers due to their intense absorption band in the visible region, their long-lived excited triplet state, and the stability of their one-electron-reduced species after electron transfer from an electron donor. In particular, [Ru(bpy)_3_]^2+^ has been widely used not only in CO_2_ reduction but also in hydrogen evolution [16] and photoredox reactions for chemical bond formations [17]. In the photochemical CO_2_ reductions using ruthenium diimine carbonyl complexes, such as [Ru(bpy)_2_(CO)(X)]^+^ (X =Cl, H), [Ru(bpy)_2_(CO)_2_]^2+^, and *cis*-Ru(bpy)(CO)_2_(Cl)_2_, formic acid is selectively formed using [Ru(bpy)_3_]^2+^ as the photosensitizer in DMF containing TEOA [18,19,20]. When 1-benzyl-1,4-dihydronicotinamide (BNAH) was used as the electron source instead of TEOA and water was used as a proton donor in DMF (DMF:water = 9:1 *v*/*v*), the photochemical CO_2_ reduction using [Ru(bpy)_2_(CO)_2_]^2+^ in the presence of [Ru(bpy)_3_]^2+^ produces CO as well as formic acid [19,20,21]. The covalently linked systems of a tris(diimine)ruthenium complex and a ruthenium diimine carbonyl complex were used in the photochemical CO_2_ reduction with BNAH as the electron donor in a mixed solvent of DMA and TEOA, yielding formic acid as the reduction product with high selectivity [22]. The photochemical CO_2_ reduction using [Ru(bpy)_2_(CO)H]^+^ in the presence of BNAH and [Ru(bpy)_3_]^2+^ in a biphasic liquid-condensed CO_2_ gas system using a mixed solvent of DMF and water produces a mixture of CO and formic acid [23].

Homogeneous photochemical reactions catalyzed by many molecular catalysts have typically been carried out in organic solvents. In the photochemical CO_2_ reduction, since Lehn et al. discovered that *fac*-Re(bpy)(CO)_3_Cl serves as a highly efficient CO_2_ reduction photocatalyst, DMF has been widely used as the solvent. However, DMF has a disadvantage of generating formic acid by hydrolysis, which complicates the distinction between this blank formic acid and the formic acid produced by the CO_2_ reduction [24]. Therefore, we proposed using *N,N*-dimethylacetamide (DMA) as an alternative solvent to DMF [25]. DMA is stable against hydrolysis and does not produce formic acid even if hydrolysis occurs. Currently, various researchers are using DMA as a solvent instead of DMF in different CO_2_ reduction systems [26,27,28,29,30,31].

On the other hand, water is an abundant and environmentally friendly solvent, making the replacement of organic solvents with water an important issue. While heterogeneous systems have achieved the photochemical CO_2_ reduction in water, where water also acts as the electron donor [32,33], attempts to replace organic solvents with water in homogeneous catalytic reactions have often been unsuccessful. We have previously reported that the photocatalytic CO_2_ reduction using [Ru(bpy)_2_(CO)_2_]^2+^, [Ru(bpy)_3_]^2+^, and BNAH produces CO and formic acid in a mixed solvent of DMA and water (DMA:water = 9:1 *v*/*v*) with high efficiency, but increasing the water ratio decreases the catalytic activity [25]. This study demonstrates that the changes in the water ratio significantly affect the initial stages of the photocatalytic reaction. Here, the concentration of the catalyst, [Ru(bpy)_2_(CO)_2_]^2+^, is 1.0 × 10^−4^ M, which is comparable to that of the photosensitizer, [Ru(bpy)_3_]^2+^ (5.0 × 10^−4^ M). Under such conditions, it is considered that the rate-determining step in the photocatalytic reaction is not the reaction on the catalyst (“catalytic cycle”) but rather the process in which the photosensitizer is reduced and the electron is transferred from the reduced photosensitizer to the catalyst (“electron relay cycle”) (vide infra) [8]. In other words, the decrease in the catalytic activity with increasing ratio of water is related to the decreased efficiency of the “electron relay cycle” process. To support this, we investigate the water dependence in a different ruthenium diimine carbonyl complex, *trans*(Cl)-[Ru(Ac-**5Bpy**-NHMe)(CO)_2_Cl_2_] (Figure 1) [34]. This complex differs in neutral initial charge, reduction potential, and catalytic activity for CO_2_ reduction compared to the divalent cationic complex, [Ru(bpy)_2_(CO)_2_]^2+^ but exhibits a similar dependence on water content. This indicates that increasing the ratio of water does not affect the catalytic reaction on the catalyst but rather the generation of the one-electron-reduced free species of [Ru(bpy)_3_]^2+^ in the “electron relay cycle”. The significant decreases in the catalytic activity cannot be explained by the decrease in the quenching efficiency of the excited state of [Ru(bpy)_3_]^2+^ by BNAH. It is further suggested that the decrease in the solvent cage-escape efficiency from the solvent cage after the electron transfer between the excited [Ru(bpy)_3_]^2+^ and BNAH is also involved. The importance of the cage-escape process from the solvent cage has recently gained attention in various photocatalytic reactions [35,36,37,38,39,40,41]. In this study, we demonstrate that the solvent composition affects the solvent cage-escape efficiency and has a substantial impact on the photocatalytic CO_2_ reduction reaction.

## 2. Results and Discussion

### 2.1. Effect of Water Content on Catalytic Activity

Photocatalytic CO_2_ reductions were conducted in DMA or DMA/water solutions using two types of the ruthenium diimine carbonyl complexes as catalysts: one was the cationic complex [Ru(bpy)_2_(CO)_2_]^2+^ [25], and the other was the neutral complex *trans*(Cl)-[Ru(Ac-**5Bpy**-NHMe)(CO)_2_Cl_2_] [34]. [Ru(bpy)_3_]^2+^ and BNAH were used as a photosensitizer and an electron donor, respectively. Figure 2 shows the relationships between the water content and the amount of the reduction products. Both plots, using [Ru(bpy)_2_(CO)_2_]^2+^ and *trans*(Cl)-[Ru(Ac-**5Bpy**-NHMe)(CO)_2_Cl_2_], show similar trends. The main reaction products are CO and formic acid resulting from the CO_2_ reduction, with negligible amounts of hydrogen regardless of the water content in the DMA. The total amount of CO and formic acid is the highest at 10 vol% water content in DMA and significantly decreases with the addition of water above 10 vol%. Moreover, the amounts in DMA without the addition of water show lower values, possibly due to a shortage of the proton source in the catalytic reaction on the ruthenium complex catalysts [42,43,44,45,46,47,48,49]. The catalytic systems described in this study are thought to consist of two main cycles: the “electron relay cycle” and the “catalytic cycle” (Figure 3) [8]. Here, relatively high catalyst concentrations of 0.10 mM are used, where the rate-determining step is thought to be the process in the “electron relay cycle” including electron transfer from BNAH to the excited photosensitizer, followed by electron transfer from the one-electron-reduced species of the photosensitizer to the catalyst (Figure 3) [50,51]. Thus, the decreases in the catalytic activities above 10 vol% water is explainable by decreases in the efficiency of the process leading to electron transfer to the ruthenium complex catalysts.

The electron–relay process begins with the formation of an encounter complex resulting from the diffusional encounter between BNAH and the excited [Ru(bpy)_3_]^2+^ (Figure 3). Within the encounter complex in a solvent cage, an electron transfer occurs to form the charge-separated encounter complex. The resulting species may diffuse out of the solvent cage to give the one-electron-reduced photosensitizer ([Ru(bpy)_3_]^+^) and BNAH^+^ or recombine to produce the ground state of [Ru(bpy)_3_]^2+^ and BNAH [8,35,51,52,53,54,55,56,57,58,59,60]. The free [Ru(bpy)_3_]^+^ that escapes from the solvent cage can provide the electron to the catalyst. We have observed the quantitative formation of BNA dimers from BNAH in the photocatalytic CO_2_ reduction [25], indicating that the BNA^•^ does not function as an electron donor and that both two electrons for the catalyst are supplied by the free [Ru(bpy)_3_]^+^. The BNA dimers are known to cause an undesired quenching process during the catalytic reaction. We previously investigated the dependence of the quenching efficiency of the excited state of [Ru(bpy)_3_]^2+^ on the water ratio in the presence of the BNA dimer [25]. The results indicated that the undesired quenching by the BNA dimer was suppressed with increasing water content, which is the opposite trend to the decrease in activity observed with a higher water ratio, as shown in Figure 2. Furthermore, since we are discussing the initial reaction rate, the contribution of the BNA dimer is considered negligible. Thus, the decline in catalytic activity observed at high water contents in Figure 2 would be attributed to a decrease in the formation efficiency of the free [Ru(bpy)_3_]^+^ by BNAH.

### 2.2. Diffusion Rate Constants of Excited [Ru(bpy)_3_]^2+^ and BNAH

The initial step in the electron relay process involves diffusional collision between the excited [Ru(bpy)_3_]^2+^ and BNAH. As shown in Figure 4, the viscosity of the binary solvent mixture comprising DMA and water increases as the water content increases from 0 to 40 vol.% [61,62]. The diffusion rate constants (*k*_diff_) between the excited [Ru(bpy)_3_]^2+^ and BNAH are calculated by Equation (1) [52], and the value of *k*_diff_ is significantly affected by the diffusion coefficient, i.e., the viscosity of the solvent.
(1)$$k_{diff}=4\pi N_{A}\left(D_{Ru}+D_{BNAH}\right)\left(r_{Ru}+r_{BNAH}\right)$$

*N_A_* is Avogadro’s number. The diffusion coefficient (*D_Ru_* and *D_BNAH_*) for the excited [Ru(bpy)_3_]^2+^ and BNAH are calculated by the Stokes–Einstein equation (*D* = *k*_B_*T*/6*πηr*, where *D* is the diffusion coefficient, *k*_B_ is Boltzmann’s constant, *T* is the absolute temperature, and *r* is the effective radius) using the effective radii of the excited [Ru(bpy)_3_]^2+^ (*r*
_Ru_ = 7.1 Å, assuming that the molecular size does not change by the excitation) [55,56,57] and BNAH (*r*
_BNAH_ ~ 6.7 Å) estimated from *r* = (*d_x_ d_y_ d_z_*)^1/3^ [63,64,65,66]. The diffusion constants and the quenching rate constants are summarized in Table 1. The quenching rate constants (*k_q_*) have been determined by the Stern–Volmer plots and the emission lifetimes of [Ru(bpy)_3_]^2+^ [25]. While the value of *k_diff_* decreases as the water content increases, the values of *k*_diff_ are one order of magnitude larger than those of *k_q_*. Thus, it is thought that the rate-determining step for *k_q_* is not the diffusional process (*k_diff_*) but the electron transfer process from BNAH to the excited [Ru(bpy)_3_]^2+^ (*k_ET_* in Figure 3).

### 2.3. Electron Transfer on the Encounter Complex

The values of *k_q_* are correlated with the diffusional effect using Equation (2) [52]
(2)$${{1/k}_{q}}^{\prime }={1/k}_{q}-1/k_{diff}$$

In Equation (2), *k_q_*’ is the rate constant for activated quenching and *k_diff_* is the diffusion rate constant. A kinetic analysis in Figure 3 gives the following relationships (Equation (3a,b)) [54,55,56,57].
(3a)$$k_{q}^{\prime }=K_{eq}k_{ET}[(k_{BET}+k_{CE})/(k_{-ET}+k_{BET}+k_{CE})]$$
(3b)$$=K_{eq}F_{et}\nu \exp\left[-\frac{\lambda }{4}{\left(1+\frac{∆G_{\mathrm{ET}}}{\lambda }\right)}^{2}/\mathrm{RT}\right]$$
where *F_et_* = (*k*_*BET*_ + *k*_*CE*_)/(*k_−ET_* + *k*_*BET*_ + *k*_*CE*_), *ν* is the frequency factor of *k*_*ET*_, ∆*G_ET_* is the free energy change of the electron transfer step, and λ is the sum of the inner- (λ_i_) and outer-sphere (λ_o_) reorganization energies. *K_eq_* is the equilibrium constant for the formation of the encounter complex (=*k_diff_/k_-diff_*). The value of *K_eq_* can be calculated from the Fuoss equation to be 6.62 M^−1^ ([Ru(bpy)_3_]^2+^ and the neutral quencher system, *d* = *r*
_Ru_ + *r*
_BNAH_ = 13.8 Å) [55,56,57]. The values of ∆*G_ET_* are determined from the Rehm–Weller equation, which is based on the half-wave reduction potential of [Ru(bpy)_3_]^2+^, the half-wave oxidation potential of BNAH, the excited state energy of [Ru(bpy)_3_]^2+^ (*E*_00_ = 2.12 eV), and a coulombic attraction term (Equation (4)) [52]. Judging from the emission peaks (Table 2), *E_00_* is expected not to be changed by the water content.
(4)$${∆G}_{ET}=E_{1/2}({BNAH}^{+}/BNAH)-E_{1/2}({Ru}^{2+}/{Ru}^{+})-E_{00}-w_{p}$$
*w_p_* is the electrostatic work necessary to bring two product ions to the close-contact distance: *w_p_* = (*Z_D+_Z_A−_*) *e*^2^/(*d ε_s_*), where *Z_D+_* and *Z_A−_* are the ion charges, and *d* is the distance between the ion centers. The values of *E*_1/2_(Ru^2+^/Ru^+^) and *E*_1/2_(BNAH^+^/BNAH) are determined by the cyclic voltammograms (CV) or the differential pulse voltammetry (DPV). The *E*_1/2_(BNAH^+^/BNAH) is estimated to be +0.14 V vs. Ag/Ag^+^, whose value does not change by increasing the water content in the solvent. On the other hand, the *E*_1/2_(Ru^2+^/Ru^+^) shows a shift from − 1.65 V vs. Ag/Ag^+^ in DMA to −1.76 V vs. Ag/Ag^+^ in DMA/water (6:4. *v*/*v*). As a result, the value of −∆*G_ET_* decreases as the water content increases (Table 2).

Meyer and coworkers reported that there are two limiting forms for *k_q_*’ in Equation (3b), where Case I is defined as *k_−ET_* << (*k_CE_* + *k_BET_*) and *F_et_* = 1 and Case II as *k_−ET_* >> (*k_CE_* + *k_BET_*) and *F_et_* = (*k_BET_* + *k*_CE_)/*k_−ET_*. They also reported that the reductive quenching proceeded via case I [54,55,56,57,67]. Then, Equation (3b) gives Equation (5).
(5)$$k_{q}^{\prime }=K_{eq}\nu \exp\left[-\frac{\lambda }{4}{\left(1+\frac{∆G_{\mathrm{ET}}}{\lambda }\right)}^{2}/\mathrm{RT}\right]$$
The logarithmic form becomes
(6)$$RT\ln k_{q}^{\prime }=RT\ln K_{eq}\nu -\frac{\lambda }{4}-\frac{∆G_{ET}}{2}\left(1+\frac{∆G_{ET}}{2\lambda }\right)$$
When |∆*G_ET_*| << 2λ, Equation (6) is simplified to Equation (7).
(7)$$RT\ln k_{q}^{\prime }=RT\ln K_{eq}\nu -\frac{\lambda }{4}-\frac{∆G_{ET}}{2}$$
Since λ =λ_i_ + λ_o_, Equation (8) then becomes
(8)$$RT\ln k_{q}^{\prime }+\frac{\lambda_{o}}{4}=RT\ln K_{eq}\nu -\frac{\lambda_{i}}{4}-\frac{∆G_{ET}}{2}$$
Equation (8) indicates that a plot of (*RT* ln *k_q_’* + λ_o_/4) vs. ∆*G_ET_* has a linear region of slope = 1/2 when λ_i_ and ν are constant. λ_o_ was calculated by Equation (9).
(9)$$\lambda_{0}(eV)=14.4∆e^{2}\left(\frac{1}{2r_{Ru}}+\frac{1}{2r_{BNAH}}-\frac{1}{r_{Ru}+r_{BNAH}}\right)\left(\frac{1}{n^{2}}-\frac{1}{\epsilon_{s}}\right)$$
where *n* and *ε_s_* are the refractive index and the static dielectric constant, respectively. The values of *w_p_*, λ_o_/4, −∆*G_ET_*, and *k_q_’* for the quenching of the excited [Ru(bpy)_3_]^2+^ by BNAH in DMA/water are summarized in Table 3.

We have observed a decrease in the quenching rate constant (*k_q_*) with increasing water content of the solvent, as observed in Table 1. Figure 5 displays a linear relationship with a slope of 1/2 in the plots of (*RT* ln *k_q_’* + λ_o_/4) vs. ∆*G_ET_*, indicating that the quenching process obeys Equation (8) and that the back-electron transfer process (*k_−ET_*) from the encounter complex to return the excited [Ru(bpy)_3_]^2+^ is negligible. The quenching rate decreases with increasing water content, resulting from a decrease in −∆*G_ET_* caused by the large negative shift of *E*_1/2_(Ru^2+^/Ru^+^) with increasing water content. Thus, the decrease in the quenching rate is one reason why the efficiency of the free [Ru(bpy)_3_]^+^ formation decreases at high water content.

### 2.4. Competition Between Cage-Escape and Back-Electron Transfer After Charge Separation

In Figure 2, the concentration of BNAH is 0.10 M. The quenching efficiency (*η_q_*) of the excited [Ru(bpy)_3_]^2+^, which is calculated from the Stern–Volmer plot [8], decreases from 98% in DMA to 81% in DMA containing 40 vol.% water (Table 4). This decrease does not fully reflect the decrease in catalytic activity observed in Figure 2, suggesting that there is another factor contributing to the decreasing in efficiency of free [Ru(bpy)_3_]^+^ formation. To obtain the free [Ru(bpy)_3_]^+^ that can provide an electron to the catalyst, the cage-escape process is required. This process competes with the back-electron transfer that occurs after charge separation in the solvent cage (Figure 3). In some cases, the cage-escape rate constant (*k_CE_*) can be theoretically determined from the Eigen equation (Equation (10)) [52,55,56,57,58,59,60].
(10)$$k_{CE}=\frac{2k_{B}T}{\pi {\left(r_{Ru}+r_{BNAH}\right)}^{3}\eta }\frac{w_{p}/RT}{1-\exp\left(w_{p}/RT\right)}$$
where *k_B_*, *R*, *T,* and *η* are the Boltzmann constant, gas constant, temperature, and solvent viscosity, respectively. *w_p_* is the electrostatic work required to bring two product ions to the close-contact distance (Table 2). The electron transfer from BNAH to the excited state of [Ru(bpy)_3_]^2+^ results in two cationic molecules which promote the cage-escape process by their electric repulsion. Increasing the water content in water/DMA solutions causes both an increase in the solvent viscosity and a decrease in the electrostatic work (*w_p_*). In particular, the increase in solvent viscosity leads to a significant decrease in the *k*_CE_ value (Figure 6a).

Table 4 summarizes the *k_CE_* values for different water ratios, the *k_CE_* values corrected by the quenching efficiency, and the relative TON values of the photocatalytic CO_2_ reduction against the TON at 10 vol% water. At 0 vol% water, the reaction on the catalyst is slow due to proton deficiency, and the rate-determining step is thought to occur in the “catalytic cycle” shown in Figure 3. In the region with more than 10 vol% water, the catalytic reaction proceeds sufficiently, and the rate-determining process is expected to occur within the “electron transfer cycle”. If the back-electron transfer rate constant (*k_BET_*) in the solvent cage remains unchanged as the water content varies, the formation efficiency of free [Ru(bpy)_3_]^+^ should be proportional to the *k_CE_* value corrected with the quenching efficiency (*η*_q_). Figure 6b shows the relationship between the relative *k_CE_* values corrected by the quenching efficiency and the relative TON values for two Ru catalysts, both of which exhibit a similar decreasing trend with increasing water content. The consistent decreasing trend observed for the two different Ru complex catalysts strongly supports that the origin of this phenomenon lies in the electron transfer cycle rather than the catalytic cycle in the region with more than 10 vol% water content. Thus, the decrease in catalytic activity with increasing water content would be significantly influenced by the efficiency of cage-escape, i.e., the efficiency of free [Ru(bpy)_3_]^+^ formation.

In the electrochemical experiments, the Faradic efficiencies of the CO_2_ reduction were estimated to be nearly 100% for the ruthenium diimine carbonyl complexes [42,43,44,45,46,47,48,49]. Considering that the electrons acquired by the catalyst are almost exclusively used for CO_2_ reduction, the relatively low photochemical quantum yield for CO and formic acid production (Φ = 15%) [25] would mainly originate from the efficiency of the electron–relay process for free [Ru(bpy)_3_]^+^ formation. Since the quenching efficiency (*η_q_*) of the excited [Ru(bpy)_3_]^2+^ is almost unity (96%) in DMA/water (9:1, *v*/*v*), the efficiency of the cage-escape, *η_CE_* = *k_CE_*/(*k_CE_* + *k_BET_*), could dominantly affect the reaction quantum yield for CO_2_ reduction. Mataga et al. reported a back-electron transfer rate constant of an order of ~10^10^ s^−1^ in the electron transfer between [Ru(bpy)_3_]^2+^ and quenchers.[70] Assuming this rate is constant, the cage-escape yield in DMA/water (9:1, *v*/*v*) is roughly estimated to be 17%, which seems to be comparable to the values of reaction quantum yield, Φ.

## 3. Materials and Methods

### 3.1. General Procedure

The ruthenium complexes, *trans*(Cl)-[Ru(Ac-**5Bpy**-NHMe)(CO)_2_Cl_2_], [Ru(bpy)_2_(CO)_2_](PF_6_)_2_, and [Ru(bpy)_3_](PF_6_)_2_, were prepared according to the literature [34,71,72]. BNAH was prepared according to the literature [73] and stored in a refrigerator. DMA (Wako, dehydrate) was used as supplied. High-purity water (resistivity: 18.2 MΩ cm) was obtained by an ultra-pure water system (RFU424TA, Advantec, Tokyo, Japan). Cyclic voltammograms (CV) and differential pulse voltmmograms (DPV) were obtained by a Bio-Logic VSP Potentiostat with a glassy-carbon working electrode (*ϕ* 3 mm), a Pt counter electrode, using tetrabutylammonium perchlorate (*^n^*Bu_4_NClO_4_) as a supporting electrolyte and a Ag/AgNO_3_ (10 mM) reference electrode in DMA and water. The quenching experiments were carried out on a F-4500 spectrometer (Hitachi, Tokyo, Japan) (λ_ex_ = 453 nm) by recording the emissions of [Ru(bpy)_3_](PF_6_)_2_ (5.8 × 10^−6^ M) in the Ar-saturated DMA/water solutions in the absence and presence of BNAH. The emission lifetimes (τ) were measured at 298 K with a FluoroCube fluorescence lifetime spectrometer (Horiba Jobin Yvon, Kyoto, Japan) using a 455 nm laser diode (NanoLED, Horiba, Kyoto, Japan).

### 3.2. Photochemical CO_2_ Reduction

Ar-saturated DMA/water solutions (5 mL) of [Ru(bpy)_2_(CO)_2_](PF_6_)_2_ (1.0 × 10^−4^ M) or *trans*(Cl)-[Ru(Ac-**5Bpy**-NHMe)(CO)_2_Cl_2_] (1.0 × 10^−4^ M), [Ru(bpy)_3_](PF_6_)_2_ (5.0 × 10^−4^ M), and BNAH (0.10 M) were placed in Quartz tubes (23 mL volume, *i.d.* = 14 mm). Each solution was bubbled through a septum cap with CO_2_ gas (1 atm) for 20 min. Ten tubes were set in a merry-go-round irradiation apparatus (Riko Kagaku, RH400-10W) and then were irradiated using a 400 W high-pressure Hg lamp at λ > 400 nm (L-39 cutoff filter, Riko Kagaku, Tokyo, Japan) at room temperature. The reaction temperature was maintained at 298 ± 3 K by using a water bath. After irradiated for a definite time, CO and H_2_ were analyzed on a system gas chromatograph (GC) based on Shimadzu GC-2014 (Shimadzu, Kyoto, Japan). The product gases (0.10 mL) were injected with a gastight syringe into the GC equipped with successive Porapak-N (GL Science, Tokyo, Japan), Molecular Sieve 13X (GL Science, Tokyo, Japan), and Shimalite-Q (Shinwa Chemical, Kyoto, Japan) columns (stainless steel columns). N_2_ (>99.9995%) was used as the carrier gas. CO was methanized through a Shimadzu MTN-1 methanizer (Shimadzu, Kyoto, Japan), followed by detection with FID (Shimadzu, Kyoto, Japan). H_2_ was detected with TCD (Shimadzu, Kyoto, Japan). Formate was extracted as formic acid with ethyl acetate prior to the GC analyses, according to the previous procedure [25]. The sample was injected into a Shimadzu GC-2014 equipped with DB-WAX (Agilent, Santa Clara, CA, USA) columns (*i.d.* 0.53 mm, 15 m × 2). Formic acid was detected with FID after methanization by a Shimadzu MTN-1 methanizer. The turnover numbers (TON) were calculated based on the amount of [Ru(bpy)_2_(CO)_2_](PF_6_)_2_ or *trans*(Cl)-[Ru(Ac-**5Bpy**-NHMe)(CO)_2_Cl_2_].

## 4. Conclusions

In this study, we systematically investigated the relationship between the water content of DMA solutions and catalytic activity for the photochemical CO_2_ reduction using [Ru(bpy)_3_]^2+^ as the photosensitizer, two types of ruthenium diimine carbonyl complexes as the catalyst, and BNAH as the electron donor. Increasing the water content led to a significant decrease in catalytic activity, which was attributed to the reduced efficiency of the electron relay cycle in the overall photocatalytic cycle (Figure 3). The water content raised the solvent viscosity, resulting in a lower diffusion rate. However, the diffusion rate constants (*k_diff_*) were an order of magnitude larger than the quenching rate constants (*k_q_*), indicating that the rate-determining step in the initial quenching process was governed by the electron transfer process in the solvent cage. The water content also affected the driving force (−∆*G_ET_*) for the electron transfer from BNAH to the excited [Ru(bpy)_3_]^2+^, where the slope of the logarithmic quenching rate constants versus the driving forces was 1/2, indicating that the back-electron transfer to form the excited [Ru(bpy)_3_]^2+^ (*k_−ET_*) was considered negligible. However, the decrease in the initial quenching process did not fully account for the significant decrease in catalytic activity. The increase in solvent viscosity also dramatically decreased the cage-escape rate (*k_CE_*), suggesting that the cage-escape efficiency was the main reason for the decrease in the efficiency of free [Ru(bpy)_3_]^+^ formation, resulting in lower catalytic activity. While the efficiency of the cage-escape process has been reported to affect catalytic activity in binary Ru(II)-Re(I) catalyst systems in water and in Ru(II) and Os(II) photosensitizers using 1,3-dimethyl-2-phenyl-2,3-dihydro-1*H*-benzo[*d*]imidazole (BIH) [36,37], systematic studies on the effects of solvents are still scarce. The present systematic investigation provides important insights not only for the reduction process of [Ru(bpy)_3_]^2+^ in photochemical CO_2_ reduction but also for replacing water in a wide range of photoredox catalytic systems involving C–C bond formation typically performed in organic solvents. In the future, the insights gained from this study on the effect of water are expected to be valuable for applications such as CO_2_ reduction reactions using water as an electron source by combining mediator molecules and water oxidation catalysts.

## Figures and Tables

**Figure 1 molecules-29-04960-f001:**
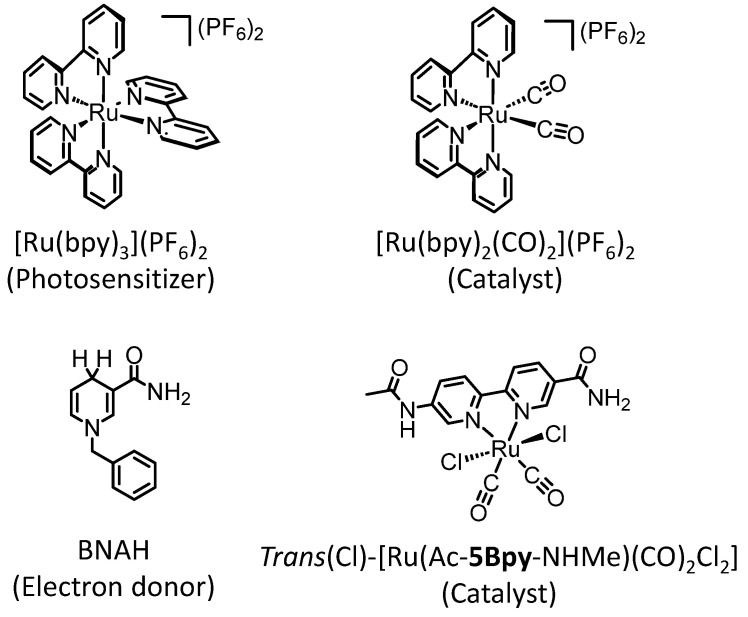
Chemical structures of [Ru(bpy)_3_]^2+^, BNAH, and ruthenium diimine carbonyl complexes.

**Figure 2 molecules-29-04960-f002:**
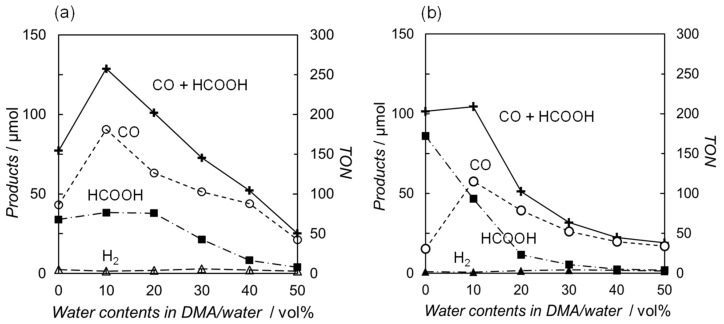
Effects of water content on the reduction products after 1h of irradiation (λ > 400 nm) using (**a**) [Ru(bpy)_2_(CO)_2_](PF_6_)_2_ (1.0 × 10^−4^ M) [25] and (**b**) *trans*(Cl)-[Ru(Ac-**5Bpy**-NHMe)(CO)_2_Cl_2_] (1.0 × 10^−4^ M) [34] in DMA containing [Ru(bpy)_3_](PF_6_)_2_ (5.0 × 10^−4^ M) and BNAH (0.10 M) under CO_2_ atmosphere: CO (○), HCOOH (■), H_2_ (Δ), and CO+HCOOH (+).

**Figure 3 molecules-29-04960-f003:**
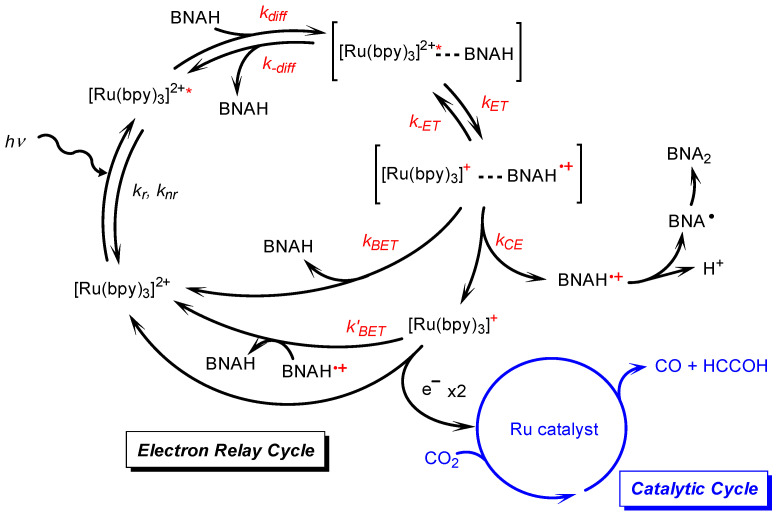
Reaction mechanism for the photocatalytic CO_2_ reduction, consisting of the electron–relay cycle and the catalytic cycle [8,51].

**Figure 4 molecules-29-04960-f004:**
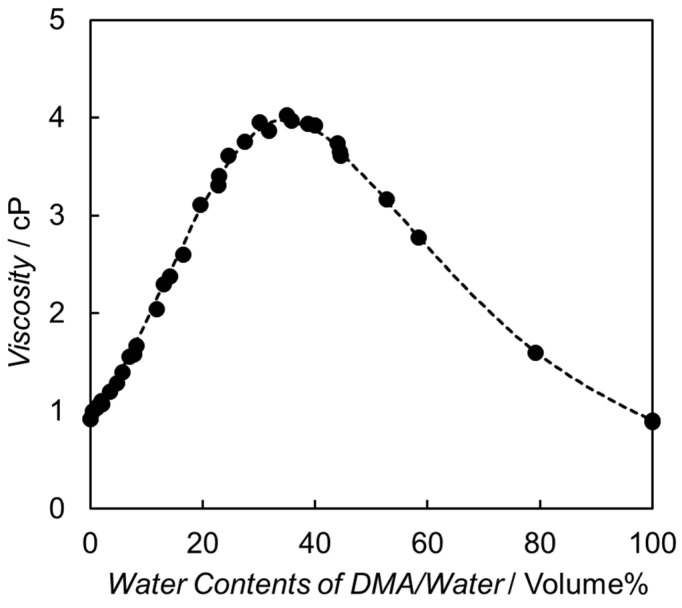
Viscosity as a function of water content in the DMA and water-mixed solvent. These plots are reconstructed using the values reported in reference [61,62].

**Figure 5 molecules-29-04960-f005:**
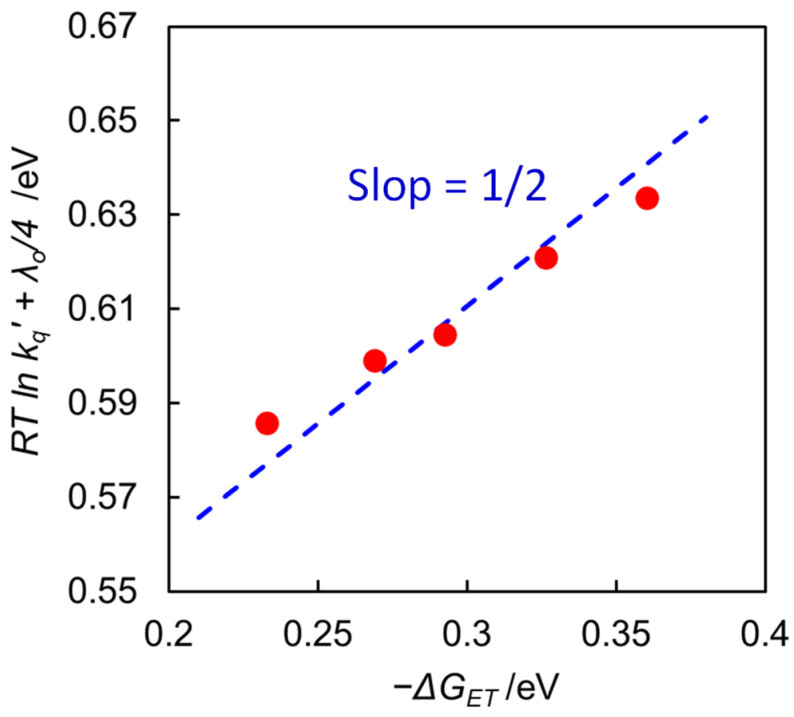
Plots of (*RT* ln *k_q_*’ + λ_o_/4) vs. ∆*G_ET_* in DMA/water at 298 K. The line is drawn with slope = 1/2.

**Figure 6 molecules-29-04960-f006:**
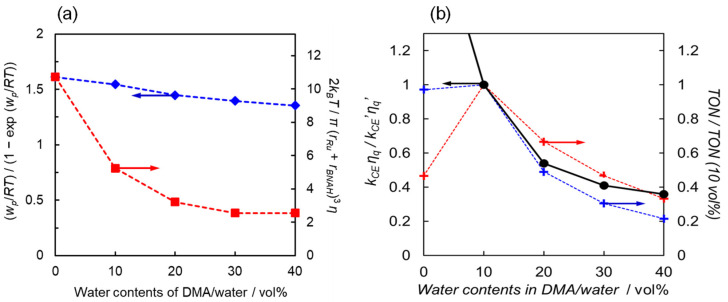
(**a**) Values of the electrostatic work term (blue diamonds) and viscosity term (red squares) in Equation (10) as a function of water content. (**b**) Relationship between the relative ratio of the cage-escape rate constants corrected by the quenching constants (black circles) and the relative TONs in the photochemical CO_2_ reduction catalysed by [Ru(bpy)_2_(CO)_2_]^2+^ (red points) and *trans*(Cl)-[Ru(Ac-**5Bpy**-NHMe)(CO)_2_Cl_2_] (blue points).

**Table 1 molecules-29-04960-t001:** Solvent effect on diffusion and quenching rate constants for excited [Ru(bpy)_3_]^2+^ and BNAH in DMA/water at 298 K.

Water Content [Vol.%]	Viscosity ^[a]^ [mPa s (=cP)]	*k_diff_* [M^−1^s^−1^]	*k_q_* ^[b]^ [M^−1^s^−1^]
0	0.93	7.1 × 10^9^	4.5 × 10^8^
10	1.9	3.5 × 10^9^	2.6 × 10^8^
20	3.1	2.1 × 10^9^	1.3 × 10^8^
30	3.9	1.7 × 10^9^	9.8 × 10^7^
40	3.9	1.7 × 10^9^	5.5 × 10^7^

^[a]^ See Figure 4. ^[b]^ Quenching rate constant estimated from the Stern–Volmer plot and the emission lifetime of [Ru(bpy)_3_]^2+^ [25].

**Table 2 molecules-29-04960-t002:** Emission peak of [Ru(bpy)_3_]^2+^, oxidation potentials of BNAH, and reduction potentials of [Ru(bpy)_3_]^2+^ in DMA and water.

Water Content [vol.%]	Emission Peak [nm] ([eV])	*E*_1/2_ ^[a]^ (BNAH^+^/BNAH) [V]	*E*_1/2_ ^[a]^ (Ru^2+^/Ru^+^) [V]	*w_p_* [eV]
0	617 (2.01)	+0.14	−1.65	0.0268
10	613 (2.00)	+0.14	−1.68	0.0238
20	615 (2.02)	+0.14	−1.71	0.0200
30	613 (2.02)	+0.14	−1.73	0.0176
40	613 (2.02)	+0.14	−1.76	0.0162

^[a]^ Redox potential vs. Ag/AgNO_3_ (0.01 M), measured in DMA/water solution containing *^n^*Bu_4_NClO_4_ (0.10 M) under Ar atmosphere.

**Table 3 molecules-29-04960-t003:** Gibbs free energy changes in electron transfer step and activation-controlled quenching rate constants in DMA and water.

Water Content [Vol.%]	*εs* ^[a]^	*n* ^[b]^	λ_o_/4 [eV]	−∆*G_ET_* [eV]	*k_q_’* [M^−1^s^−1^]
0	38.9	1.44	0.120	0.307	4.8 × 10^8^
10	43.9	1.43	0.121	0.279	2.8 × 10^8^
20	52.3	1.43	0.123	0.253	1.4 × 10^8^
30	59.3	1.42	0.125	0.234	1.0 × 10^8^
40	64.3	1.41	0.127	0.200	5.7 × 10^7^

^[a]^ Static dielectric constant calculated from the literature [68]. ^[b]^ Refractive index [69].

**Table 4 molecules-29-04960-t004:** Effect of water content on quenching efficiency, cage-escape rate constant, and TON in Figure 2.

Water Content [vol.%]	*η_q_* ^[a]^	*k_CE_* [s^−1^]	*k_CE_ η_q_/k_CE_’ η_q_’* ^[b]^	TON ^[c]^	
				[Ru(bpy)_2_(CO)_2_]^2+^	*Trans*(Cl)-[Ru(Ac-5Bpy-NHMe)(CO)_2_Cl_2_]
0	0.98	1.7 × 10^9^	2.2	154 (0.59)	203 (0.97)
10	0.96	8.0 × 10^8^	1	258 (1)	209 (1)
20	0.91	4.6 × 10^8^	0.54	202 (0.78)	102 (0.49)
30	0.89	3.5 × 10^8^	0.41	145 (0.56)	64 (0.31)
40	0.81	3.4 × 10^8^	0.36	104 (0.40)	45 (0.22)

^[a]^ Quenching fraction of emission from [Ru(bpy)_3_]^2+^ in the presence of 0.1 M BNAH, calculated as 0.1 *K_SV_*/(1 + 0.1 *K_SV_*). ^[b]^ *k_CE_’* and *η_q_’* are the values at 10 vol.% water. ^[c]^ The values in parentheses are the ratio to the TON at 10 vol% water.

## Data Availability

Further inquiries can be directed to the corresponding author/s.

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
