# Peer review of "Exploring the Impact of Water Content in Solvent Systems on Photochemical CO2 Reduction Catalyzed by Ruthenium Complexes"

_molecules, 2024, doi:10.3390/molecules29204960_

Round 1

Reviewer 1 Report

Comments and Suggestions for Authors

In this paper, the catalysis effects of two ruthenium complexes on CO2 photoreduction activity were evaluated in DMA solvent with different water content. With some derived formula, their experiment data and some experimental data previously reported, which are quite comprehensive, were analyzed. The overall conclusion is quite interesting. It should be published in our journal. However, before accepted, the manuscript still requires some revision work:

1. The author mentioned that the two selected ruthenium complexes,“trans(Cl)-[Ru(Ac-5Bpy-NHMe)(CO)2Cl2] and [Ru(bpy)2(CO)2](PF6)2”, have different initial charges. Do these two Ru atoms have the same valence states? In general, what is the reason for the significantly different effects on the photocatalytic reduction efficiency of this same active metal atom? With different charges or valence states or other reasons? Please explain the possible reasons for choosing these two contrast catalysts.

2. It is well known that photoreduction of CO2 to CO or formic acid is an electron transfer process. In Figure 2, we see that the selectivity of the two catalysts to produce CO or formic acid is not the same without water added. In particular, Figure 2 (b) shows a higher selectivity for formic acid formation without water for trans(Cl)-[Ru(Ac-5Bpy-NHMe)(CO)2Cl2] catalyst, while the selectivity/content of formic acid production decreases with the increase of water content. This seems inconsistent with the viewpoint of the authors in the paper--possibly due to shortage of the proton source in the catalytic reaction on the ruthenium complex catalysts.Page 4)”. Please explain this issue.

3. In previous work, it is acquired “the quantitative formation of BNA dimers from BNAH” (on Page 5). Based upon this fact, the author believes that "BNA• does not function as the electron donor”. Then, how to explain that the variation trend of the quantity of BNA dimers produced (which is unchanged as reported in your previous work?) with increasing water content?

4. On Page 8, please check the equation (10): "exp(wp/RT)" misses “w”.

5. In Table 4 on Page 9, the authors listed the quenching efficiency as a function of the water content, cage-escape rate constant, and TON, etc. The results derived from the formula indicate that both quenching efficiency and cage-escape rate decrease as water content increases. However, the experimental results show that TON initially increases and then decreases with increasing water content, reaching a maximum at 10 vol.%. The authors should discuss whether including the 0 vol.% water content in the comparative analysis is appropriate, as adding water up to 10 vol.% can also increase solvent viscosity, but promote the activity. These need to be discussed in details in the paper.

Comments on the Quality of English Language

Good

Author Response

 In this paper, the catalysis effects of two ruthenium complexes on CO2 photoreduction activity were evaluated in DMA solvent with different water content. With some derived formula, their experiment data and some experimental data previously reported, which are quite comprehensive, were analyzed. The overall conclusion is quite interesting. It should be published in our journal. However, before accepted, the manuscript still requires some revision work:

→ We appreciate the reviewer's high evaluation of our manuscript.

  1. The author mentioned that the two selected ruthenium complexes, “trans(Cl)-[Ru(Ac-5Bpy-NHMe)(CO)2Cl2] and [Ru(bpy)2(CO)2](PF6)2”, have different initial charges. Do these two Ru atoms have the same valence states? In general, what is the reason for the significantly different effects on the photocatalytic reduction efficiency of this same active metal atom? With different charges or valence states or other reasons? Please explain the possible reasons for choosing these two contrast catalysts.

→ Although the initial charge of the Ru ion is considered to be the same as Ru(II), the overall charge and reduction potential are different, leading to different CO₂ reduction activities as Ru complex catalysts. In photocatalytic reactions using a photosensitizer, the photosensitizer becomes the reduced form, and the electron transfer from this reduced species to the catalyst also influences the reaction rate. In the experiments, despite the use of two different catalysts, there was no significant change in the amount of CO₂ reduction products, suggesting that the rate-determining step of the entire catalytic reaction lies in the electron transfer process from the photosensitizer to the catalyst (Coord. Chem. Rev. 2018, 373, 333). This process is greatly affected by the water ratio. In the introduction, the following explanation was added about the process of photocatalytic reactions using a photosensitizer, as well as the rationale for using two different Ru complex catalysts.

 “ Here, the concentration of the catalyst, [Ru(bpy)2(CO)2]2+, is 1.0 × 10–4 M, which is comparable to that of the photosensitizer, [Ru(bpy)3]2+ (5.0 × 10–4 M). Under such conditions, it is considered that the rate-determining step in the photocatalytic reaction is not the reaction on the catalyst (“catalytic cycle”), but rather the process in which the photosensitizer is reduced, and the electron is transferred from the reduced photosensitizer to the catalyst (electron relay cycle”) (vide infra).[1h] In other words, the decrease in the catalytic activity with increasing ratio of water is related to the decreased efficiency of the electron relay cycle” process.”

  1. It is well known that photoreduction of CO2 to CO or formic acid is an electron transfer process. In Figure 2, we see that the selectivity of the two catalysts to produce CO or formic acid is not the same without water added. In particular, Figure 2 (b) shows a higher selectivity for formic acid formation without water for trans(Cl)-[Ru(Ac-5Bpy-NHMe)(CO)2Cl2] catalyst, while the selectivity/content of formic acid production decreases with the increase of water content. This seems inconsistent with the viewpoint of the authors in the paper--“possibly due to shortage of the proton source in the catalytic reaction on the ruthenium complex catalysts.(Page 4)”. Please explain this issue.

→ The overall production of CO and formic acid decreased compared to the amount predicted in Figure 6b. This decrease is likely due to the requirement for protons in the catalytic process, which may not have proceeded efficiently. The high selectivity for formic acid production is consistent with the previous report in electrochemical CO2 reduction reactions (reference 20 in the revised manuscript), where we observed that decreasing the proton concentration led to an increase in formic acid production. It is thought that a similar phenomenon occurred in the reaction using trans(Cl)-[Ru(Ac-5Bpy-NHMe)(CO)2Cl2] without the addition of water.

  1. In previous work, it is acquired “the quantitative formation of BNA dimers from BNAH” (on Page 5). Based upon this fact, the author believes that "BNA• does not function as the electron donor”. Then, how to explain that the variation trend of the quantity of BNA dimers produced (which is unchanged as reported in your previous work?) with increasing water content?

→ We have previously investigated the water ratio dependence of the quenching efficiency of the excited state of [Ru(bpy)3]2+ by BNA dimers (reference 13 in the revised manuscript). The results indicate that increasing the water content decreases the quenching processes caused by BNA dimers, which shows a trend opposite to the decrease in quenching efficiency associated with the increased water ratio. Moreover, since we are discussing the initial rates, the contribution of BNA dimers is likely to be minimal.

The following sentence has been added to provide the additional explanation:

The BNA dimers are known to cause an undesired quenching process during the catalytic reaction. We previously investigated the dependence of the quenching efficiency of the excited state of [Ru(bpy)₃]²⁺ on the water ratio in the presence of the BNA dimer.[13] The results indicated that the undesired quenching by the BNA dimer was suppressed with increasing water content, which is the opposite trend to the decrease in activity observed with a higher water ratio, as shown in Figure 2. Furthermore, since we are discussing the initial reaction rate, the contribution of the BNA dimer is considered negligible.

  1. On Page 8, please check the equation (10): "exp(wp/RT)" misses “w”.

→ It seems to have disappeared when converting to PDF file. We have corrected it.

  1. In Table 4 on Page 9, the authors listed the quenching efficiency as a function of the water content, cage-escape rate constant, and TON, etc. The results derived from the formula indicate that both quenching efficiency and cage-escape rate decrease as water content increases. However, the experimental results show that TON initially increases and then decreases with increasing water content, reaching a maximum at 10 vol.%. The authors should discuss whether including the 0 vol.% water content in the comparative analysis is appropriate, as adding water up to 10 vol.% can also increase solvent viscosity, but promote the activity. These need to be discussed in details in the paper.

→ In the region where the water content is 0 vol%, the rate-determining step is expected to be on the "catalytic cycle" shown in Figure 3. In contrast, in regions with more than 10 vol% water content, the rate-determining step is anticipated to be part of the "electron relay cycle." To clarify this distinction and emphasize that we are discussing region with more than 10 vol% water content, we have added the following sentence and the phrase "in region with more than 10 vol% water content"

At 0 vol% water, the reaction on the catalyst is slow due to proton deficiency, and the rate-determining step is thought to occur in the “catalytic cycle” shown in Figure 3. In the region with more than 10 vol% water, the catalytic reaction proceeds sufficiently, and the rate-determining process is expected to occur within the “electron transfer cycle.

Reviewer 2 Report

Comments and Suggestions for Authors

Paper fits the scope of the journal. However its novelty and interest for readers is rather moderate. Novetly should be clearly shown. Authors used 37 referenceces. Most of them are not recent. Application area of the system should be disscused in reference to obtained process efficiency.

Author Response

Paper fits the scope of the journal. However its novelty and interest for readers is rather moderate. Novetly should be clearly shown. Authors used 37 referenceces. Most of them are not recent. Application area of the system should be disscused in reference to obtained process efficiency.

→ According to the reviewer’s comments, we have cited recent works and included the significance of in the Introduction:

The added citations:

[19a] Draper, F.; DiLuzio, S.; Sayre, H. J.; Pham, L. N.; Coote, M. L.; Doeven, E. H.; Francis, P. S.; Connell, T. U. Maximizing Photon-to-Electron Conversion for Atom Efficient Photoredox Catalysis. J. Am. Chem. Soc. 2024, 146, 26830–26843.

[19b] Ripak, A.; Vega Salgado, A. K.; Valverde, D.; Cristofaro, S.; de Gary, A.; Olivier, Y.; Elias, B.; Troian-Gautier, L. Factors Controlling Cage Escape Yields of Closed- and Open-Shell Metal Complexes in Bimolecular Photoinduced Electron Transfer. J. Am. Chem. Soc. 2024, 146, 22818–22828.

[19c] De Kreijger, S.; Ripak, A.; Elias, B.; Troian-Gautier, L. Investigation of the Excited-State Electron Transfer and Cage Escape Yields Between Halides and a Fe(III) Photosensitizer. J. Am. Chem. Soc. 2024, 146, 10286–10292.

[19d] Wang, C.; Li, H.; Bürgin, T. H.; Wenger, O. S. Cage Escape Governs Photoredox Reaction Rates and Quantum Yields. Nat. Chem. 2024, 16, 1151– 1159.

The sentences added at the end of the introduction:

“The importance of the cage-escape process from the solvent cage has recently gained attention in various photocatalytic reactions.[17-23] In this study, we demonstrate that the solvent composition affects the solvent cage-escape efficiency and has a substantial impact on the photocatalytic CO₂ reduction reaction.”

Reviewer 3 Report

Comments and Suggestions for Authors

What this article looks at is how the amount of water affects the ability of ruthenium complexes to reduce CO2 in a system that includes water and N,N-dimethylacetamide (DMA).

The findings indicate that elevated water content markedly diminishes CO and fatty acid production, which correlates with reduced efficiency in the generation of reduced rutenium species. The study shows that water concentration does not directly influence catalysis; instead, it impacts the electron transfer process.

1. This article discusses a significant subject in the quest for alternative solvents in CO2 photocatalysis. Nonetheless, there are aspects that could be enhanced, including experimental emphasis, the profundity of comparative analysis, and the clarity of hypotheses. 
Revise the initial section to clarify the primary hypothesis and research objectives.

2. The analysis of the study regarding the influence of water content relies on only two materials. Incorporating other varieties of transition metal complexes for comparative analysis would enhance the robustness of the conclusions (or materials reported in the bibliography).

3. The work underscores essential chemistry yet neglects to consider the practical implications of CO2 reduction. Incorporate a segment addressing the application of these findings in industrial or environmental contexts, emphasizing the scalability of the method for actual implementation.

4-Figures. Enhance clarity by adjusting the graph.

For instance, in Figure 6, the graphs might be oriented vertically to merge and share the X-axis of water content, so eliminating confusion with the Y-axes.

5: While the mechanical routes are outlined, there is less discussion on the possible influence of environmental conditions (e.g., temperature or pressure) on these pathways.

Provide a thorough analysis of the possible alterations to the reaction process that may arise due to varying experimental conditions, like temperature or CO2 pressure.

Author Response

The findings indicate that elevated water content markedly diminishes CO and fatty acid production, which correlates with reduced efficiency in the generation of reduced rutenium species. The study shows that water concentration does not directly influence catalysis; instead, it impacts the electron transfer process.

1. This article discusses a significant subject in the quest for alternative solvents in CO2 photocatalysis. Nonetheless, there are aspects that could be enhanced, including experimental emphasis, the profundity of comparative analysis, and the clarity of hypotheses. Revise the initial section to clarify the primary hypothesis and research objectives.
→The following sentences were added to the Introduction to clarify the significance and purpose of this study.

“Here, the concentration of the catalyst, [Ru(bpy)2(CO)2]2+, is 1.0 × 10–4 M, which is comparable to that of the photosensitizer, [Ru(bpy)3]2+ (5.0 × 10–4 M). Under such conditions, it is considered that the rate-determining step in the photocatalytic reaction is not the reaction on the catalyst (“catalytic cycle”), but rather the process in which the photosensitizer is reduced, and the electron is transferred from the reduced photosensitizer to the catalyst (“electron relay cycle”) (vide infra).[1h] In other words, the decrease in the catalytic activity with increasing ratio of water is related to the decreased efficiency of the “electron relay cycle” process.”

and

“The importance of the cage-escape process from the solvent cage has recently gained attention in various photocatalytic reactions.[17-19] In this study, we demonstrate that the solvent composition affects the solvent cage-escape efficiency and has a substantial impact on the photocatalytic CO₂ reduction reaction.”

2. The analysis of the study regarding the influence of water content relies on only two materials. Incorporating other varieties of transition metal complexes for comparative analysis would enhance the robustness of the conclusions (or materials reported in the bibliography).
→Unfortunately, we have not been able to conduct experiments with other metal complexes. However, we believe that linking the cage escape efficiency to the products of the CO₂ reduction reaction is a significant advancement in this field. Furthermore, since systematic studies on the effects of solvents are still scarce, we believe that our research will serve as a valuable reference for various other photocatalytic redox reactions.

3. The work underscores essential chemistry yet neglects to consider the practical implications of CO2 reduction. Incorporate a segment addressing the application of these findings in industrial or environmental contexts, emphasizing the scalability of the method for actual implementation.
→The following sentence was added to the Conclusion to emphasize the future application:
“In the future, the insights gained from this study on the effect of water are expected to be valuable for applications such as CO₂ reduction reactions using water as an electron source, by combining mediator molecules and water oxidation catalysts.”

4. Figures. Enhance clarity by adjusting the graph. For instance, in Figure 6, the graphs might be oriented vertically to merge and share the X-axis of water content, so eliminating confusion with the Y-axes.

→We have adjusted the figures to an appropriate size. As for Figure 6, while we attempted to implement the reviewer’s suggestions, the plots became overlapped and overly complex. Additionally, it became difficult to correlate the decrease in TON as clearly. Therefore, we have chosen to maintain the original format.

5. While the mechanical routes are outlined, there is less discussion on the possible influence of environmental conditions (e.g., temperature or pressure) on these pathways. Provide a thorough analysis of the possible alterations to the reaction process that may arise due to varying experimental conditions, like temperature or CO2 pressure.
→We have not yet controlled for factors such as temperature and CO₂ pressure. As commented by the reviewer, we agree that investigating various conditions will be an important direction for future work.
We have now added the missing details regarding the temperature during light irradiation and the CO₂ pressure in the Materials and Methods.

Round 2

Reviewer 3 Report

Comments and Suggestions for Authors

The authors have addressed the marked recommendations, improving the quality of the work.

In the section: Photochemical CO2 reduction., is atom correct? or does it refer to atm?

Author Response

In the section: Photochemical CO2 reduction., is atom correct? or does it refer to atm?

>Thank you for your careful review. We corrected it.